# Collaborative Learning for Deep Neural Networks

**Guocong Song**
Playground Global
Palo Alto, CA 94306
songgc@gmail.com

**Wei Chai**
Google
Mountain View, CA 94043
chaiwei@google.com

## Abstract

We introduce collaborative learning in which multiple classifier heads of the same network are simultaneously trained on the same training data to improve generalization and robustness to label noise with no extra inference cost. It acquires the strengths from auxiliary training, multi-task learning and knowledge distillation. There are two important mechanisms involved in collaborative learning. First, the consensus of multiple views from different classifier heads on the same example provides supplementary information as well as regularization to each classifier, thereby improving generalization. Second, *intermediate-level representation* (ILR) sharing with backpropagation rescaling aggregates the gradient flows from all heads, which not only reduces training computational complexity, but also facilitates supervision to the shared layers. The empirical results on CIFAR and ImageNet datasets demonstrate that deep neural networks learned as a group in a collaborative way significantly reduce the generalization error and increase the robustness to label noise.

## 1 Introduction

When training deep neural networks, we must confront the challenges of general nonconvex optimization problems. Local gradient descent methods that most deep learning systems rely on, such as variants of *stochastic gradient descent* (SGD), have no guarantee that the optimization algorithm will converge to a global minimum. It is well known that an ensemble of multiple instances of a target neural network trained with different random seeds generally yields better predictions than a single trained instance. However, an ensemble of models is too computationally expensive at inference time. To keep the exact same computational complexity for inference, several training techniques have been developed by adding additional networks in the training graph to boost accuracy without affecting the inference graph, including auxiliary training [19], multi-task learning [4, 3], and knowledge distillation [10]. Auxiliary training is introduced to improve the convergence of deep networks by adding auxiliary classifiers connected to certain intermediate layers [19]. However, auxiliary classifiers require specific new designs for their network structures in addition to the target network. Furthermore, it is found later [20] that auxiliary classifiers do not result in obvious improved convergence or accuracy. Multi-task learning is an approach to learn multiple related tasks simultaneously so that knowledge obtained from each task can be reused by the others [4, 3, 21]. However, it is not useful for a single task use case. Knowledge distillation is introduced to facilitate training a smaller network by transferring knowledge from another high-capacity model, so that the smaller one obtains better performance than that trained by using labels only [10]. However, distillation is not an end-to-end solution due to having two separate training phases, which consume more training time.

In this paper, we propose a framework of collaborative learning that trains several classifier heads of the same network simultaneously on the same training data to cope with the above challenges. The method acquires the advantages from auxiliary training, multi-task learning, and knowledge distillation, such as, appending the exact same network as the target one in the training graph for a

single task, sharing intermediate-level representation (ILR), learning from the outputs of other heads (peers) besides the ground-truth labels, and keeping the inference graph unchanged. Experiments have been performed with several popular deep neural networks on different datasets to benchmark performance, and their results demonstrate that collaborative learning provides significant accuracy improvement for image classification problems in a generic way. There are two major mechanisms collaborative learning benefits from: 1) The consensus of multiple views from different classifier heads on the same data provides supplementary information and regularization to each classifier. 2) Besides computational complexity reduction benefited from ILR sharing, backpropagation rescaling aggregates the gradient flows from all heads in a balanced way, which leads to additional performance enhancement. The per-layer network weight distribution shows that ILR sharing reduces the number of "dead" filter weights in the bottom layers due to the vanishing gradient issue, thereby enlarging the network capacity.

The major contributions are summarized as follows. 1) Collaborative learning provides a new training framework that for any given model architecture, we can use the proposed collaborative training method to potentially improve accuracy, with no extra inference cost, with no need to design another model architecture, with minimal hyperparameter re-tuning. 2) We introduce ILR sharing into co-distillation that not only enhances training time/memory efficiency but also improves generalization error. 3) Backpropagation rescaling we propose to avoid gradient explosion when the number of heads is big is also proven able to improve accuracy when the number of heads is small. 4) Collaborative learning is demonstrated to be robust to label noise.

## 2 Related work

In addition to auxiliary training, multi-task learning, and distillation mentioned before, we list other related work as follows.

**General label smoothing.** Label smoothing replaces the hard values (1 or 0) in one-hot labels for a classifier with smoothed values, and is shown to reduce the vulnerability of noisy or incorrect labels in datasets [20]. It regularizes the model and relaxes the confidence on the labels. Temporal ensembling forms a consensus prediction of the unknown labels using the outputs of the network-in-training on different epochs to improve the performance of semi-supervised learning [14]. However, it is hard to scale for a large dataset since temporal ensembling requires to memorize the smoothed label of each data example.

**Two-way distillation.** Co-distillation of two instances of the same neural network is studied in [2] with a focus on training speed-up in a distributed learning environment. Two-way distillation between two networks, which can use the same architecture or different, is also studied in [23]. Each of them alternatively optimizes its own network parameters. However, the developed algorithms are far from optimized. First, when different classifiers have different architectures, each of them should have a different weight associated with its loss function to balance injected backpropagation error flows. Second, multiple copies of the target network increase proportionally the memory consumption in *graphics processing unit* (GPU) and the training time.

**Self-distillation/born-again neural networks.** Self-distillation is a kind of distillation when the student network is identical to the teacher in terms of the network graph. Furthermore, the distillation process can be performed consecutively several times. At each consecutive step, a new identical model is initialized from a different random seed and trained from the supervision of the earlier generation. At the end of the procedure, additional gains can be achieved with an ensemble of multiple students generations [7]. However, multiple self-distillation processes multiply the total training time proportionally; an ensemble of multiple student generations increases the inference time accordingly as well.

In comparison, the major goal of this paper is to improve the accuracy of a target network without changing its inference graph and emphasize both the accuracy and the training efficiency.

## 3 Collaborative learning

The framework of collaborative learning consists of three major parts: the generation of a population of classifier heads in the training graph, the formulation of the learning objective, and optimization

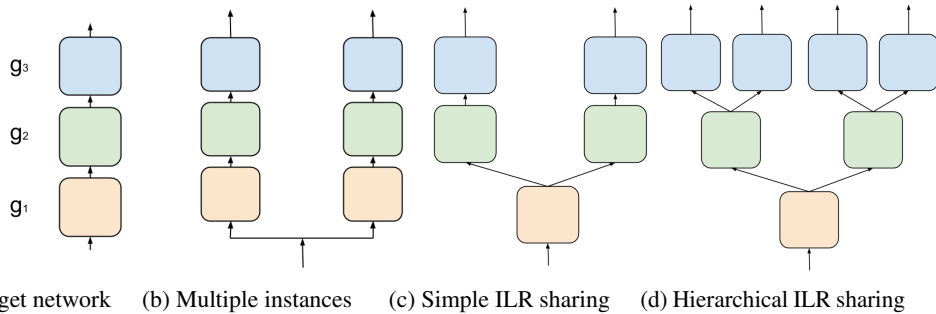

| (a) Target network | (b) Multiple instances | (c) Simple ILR sharing | (d) Hierarchical ILR sharing |

Figure 1: **Multiple head patterns for training.** Three colors represent subnets $g_1, g_2$, and $g_3$ in (1).

for learning a group of classifiers collaboratively. We will describe the details of each of them in the following subsections.

## 3.1  Generation of training graph

Similar to auxiliary training [19], we add several new classifier heads into the original network graph during training time. At inference time, only the original network is kept and all added parts are discarded. Unlike auxiliary training, each classifier head here has an identical network to the original one in terms of graph structure. This approach leads to advantages over auxiliary training in terms of engineering effort minimization. First, it does not require to design additional networks for the auxiliary classifiers. Second, the structure symmetry for all heads does not require additional different weights associated with loss functions to well balance injected backpropagation error flows, because an equal weight for each head's objective is optimal for training.

Figure 1 illustrates several patterns to create a group of classifiers in the training graph. Figure 1 (a) is a target network to train. The network can be expressed as $z = g(x; \theta)$, where $g$ is determined by the graph architecture, and $\theta$ represents the network parameters. To better explain the following patterns, we assume the network $g$ can be represented as a cascade of three functions or subnets,

$$g(x; \theta) = g_3(g_2(g_1(x; \theta_1); \theta_2); \theta_3) \tag{1}$$

where $\theta = [\theta_1, \theta_2, \theta_3]$ and $\theta_i$ includes all parameters of subnet $g_i$ accordingly. In Figure 1 (b), each head is just a new instance of the original network. The output of head $h$ is $z^{(h)} = g(x; \theta^{(h)})$, where $\theta^{(h)}$ is an instance of network parameters for head $h$. Another pattern allows all heads to share ILRs in the same low layers, which is shown in Figure 1 (c). This structure is very similar to multi-task learning [4, 3], in which different supervised tasks share the same input, as well as some ILR. However, collaborative learning has the same supervised tasks for all heads. It can be expressed as follows $z^{(h)} = g_3(g_2(g_1(x; \theta_1); \theta_2^{(h)}); \theta_3^{(h)})$, where there is only one instance of $\theta_1$ shared by all heads. Furthermore, multi-heads can take advantage of multiple hierarchical ILRs, as shown in Figure 1 (d). The hierarchy is similar to a binary tree in which the branches at the same levels are copies of each other. For inference, we just need to keep one head with its dependent nodes and discard the rest. Therefore, the inference graph is identical to the original graph $g$.

It is shown in [17, 5] that the training memory size is roughly proportional to the number of layers/operations. With the multi-instance pattern, the number of parameters in the whole training graph is proportional to the number of heads. Obviously, ILR sharing can proportionally reduce the memory consumption and speed up training, compared to multiple instances without sharing. It is more interesting that the empirical results and analysis in Section 4 will demonstrate that ILR sharing is able to boost the classification accuracy as well.

## 3.2  Learning objectives

The main idea of collaborative learning is that each head learns from ground-truth labels but also from the whole population through the training process. We focus on multi-class classification problems in this paper. For head $h$, the classifier's logit vector is represented as $z = [z_1, z_2, \ldots, z_m]^{tr}$ for $m$

classes. The associated softmax with temperature $T$ is defined as follows,

$$\sigma_i(\boldsymbol{z}^{(h)}; T) = \frac{\exp\left(z_i^{(h)}/T\right)}{\sum\limits_{j=1}^{m} \exp\left(z_j^{(h)}/T\right)} \tag{2}$$

When $T = 1$, (2) is just a normal softmax function. Using a higher value for $T$ produces a softer probability distribution over classes. The loss function for head $h$ is proposed as

$$L^{(h)} = \beta J_{hard}(\boldsymbol{y}, \boldsymbol{z}^{(h)}) + (1 - \beta)J_{soft}(\boldsymbol{q}^{(h)}, \boldsymbol{z}^{(h)}) \tag{3}$$

where $\beta \in (0, 1]$. The objective function with regard to a ground-truth label $J_{hard}$ is just the classification loss – cross entropy between a one-hot encoding of the label $\boldsymbol{y}$ and the softmax output with temperature of 1: $J_{hard}(\boldsymbol{y}, \boldsymbol{z}^{(h)}) = -\sum_{i=1}^{m} y_i \log(\sigma_i(\boldsymbol{z}^{(h)}; 1))$. The soft label of head $h$ is proposed to be a consensus of all other heads' predictions as follows:

$$\boldsymbol{q}^{(h)} = \sigma\left(\frac{1}{H-1}\sum_{j \neq h} \boldsymbol{z}^{(j)}; T\right)$$

which combines the multiple views on the same data and contains additional information rather than the ground-truth label. The objective function with regard to the soft label is the cross entropy between the soft label and the softmax output with a certain temperature, i.e.

$$J_{soft}(\boldsymbol{q}^{(h)}, \boldsymbol{z}^{(h)}) = -\sum_{i=1}^{m} q_i^{(h)} \log(\sigma_i(\boldsymbol{z}^{(h)}; T))$$

which can be regarded as a distance measure between an average prediction from population and the prediction of each head [10]. Minimizing this objective aims at transferring the information from the soft label to the logits and regularizing the training network.

### 3.3 Optimization for a group of classifier heads

In addition to performance optimization, another design criterion for collaborative learning is to keep the hyperparameters in training algorithms, e.g. the type of SGD, regularization, and learning rate schedule, the same as those used in individual learning. Thus, collaborative learning can be simply put on top of individual learning. The optimization here is mainly designed to take new concepts involved in collaborative learning into account, including a group of classifiers, and ILR sharing.

**Simultaneous SGD.** Since multiple heads are involved in optimization, it seems straightforward to alternatively update the parameters associated with each head one-by-one. This algorithm is used in both [23, 2]. In fact, alternative optimization is popular in generative adversarial networks [8], in which a generator and discriminator get alternatively updated. However, alternative optimization has the following shortcomings. In terms of speed, it is slow because one head needs to recalculate a new prediction after updating its parameters. In terms of convergence, recent work [15, 16] reveals that simultaneous SGD has faster convergence and achieves better performance than the alternative one. Therefore, we propose to apply SGD and update all parameters simultaneously in the training graph according to the total loss, which is the sum of each head's loss as well as regularization $\Omega(\boldsymbol{\theta})$.

$$L = \sum_{h=1}^{H} L^{(h)} + \lambda\Omega(\boldsymbol{\theta}) \tag{4}$$

We suggest keeping the same regularization and its hyperparameters as individual training when applying collaborative learning. It is important to avoid unnecessary hyperparameter search in practice when introducing a new training approach. The effectiveness of simultaneous SGD will be validated in Section 4.1.

**Backpropagation rescaling.** First, we describe an important stability issue with ILR sharing. Assume that there are $H$ heads sharing subnet $g_1(\cdot; \boldsymbol{\theta}_1)$ as shown in Figure 2 (a), in which $\boldsymbol{\theta}_1$ and $\boldsymbol{\theta}_2^{(h)}$ represent the parameters of $g_1$ and those of $g_2$ associated with head $h$, respectively. The output of

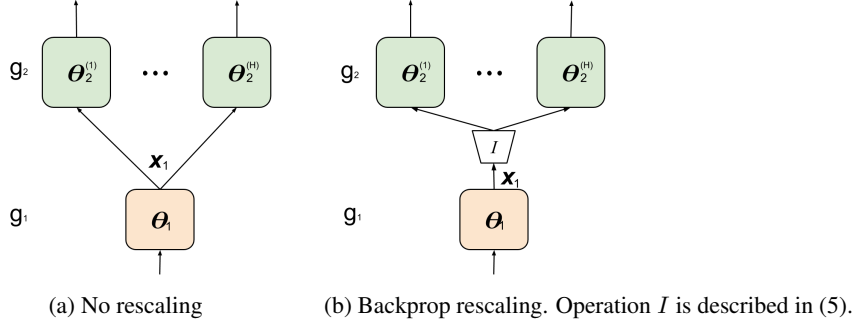

(a) No rescaling  (b) Backprop rescaling. Operation $I$ is described in (5).

Figure 2: No rescaling vs backpropagation rescaling

the shared layers, $x_1$, is fed to all corresponding heads. However, the backward graph becomes a many-to-one connection. According to (4), the backpropagation input for the shared layers is $\nabla_{x_1} L = \sum_{h=1}^{H} \nabla_{x_1} L^{(h)}$. It is not hard to discover an issue that the variance of $\nabla_{x_1} L$ grows as the number of heads grows. Assume that the gradient of each head's loss has a limited variance, i.e., $\text{Var}((\nabla_{x_1} L^{(h)})_i) < \infty$, where $i$ represents each element in a vector. We should make the system stable, i.e., $\text{Var}((\nabla_{x_1} L)_i) < \infty$, even when $H \to \infty$. Unfortunately, the backpropagation flow of Figure 2 (a) is unstable in the asymptotic sense due to the sum of all gradient flows.

Note that simple loss scaling, i.e., $L = \frac{1}{H} \sum_h L^{(h)}$, bring another problem: resulting in very slow learning w.r.t $\theta_2^{(h)}$. The SGD update is $\theta_2^{(h)} \leftarrow \theta_2^{(h)} - \eta \frac{1}{H} \nabla_{\theta_2^{(h)}} L^{(h)}$. For a fixed learning rate $\eta$, $\eta \frac{1}{H} \nabla_{\theta_2^{(h)}} L^{(h)} \to 0$ when $H \to \infty$.

Therefore, backpropagation rescaling is proposed to achieve two goals at the same time – to normalize the backpropagation flow in subnet $g_1$ and keep that in subnet $g_2$ the same as the single classifier case. The solution to add a new operation $I(\cdot)$ between $g_1$ and $g_2$, shown in Figure 2 (b), which is

$$I(x) = x, \quad \nabla_x I = \frac{1}{H} \tag{5}$$

And then the backpropagation input for the shared layers becomes

$$\nabla_{x_1} L = \frac{1}{H} \sum_{h=1}^{H} \nabla_{x_1} L^{(h)} \tag{6}$$

The variance of (6) is then always limited, which is proven in Session 1 of Supplementary material. Backpropagation rescaling is essential for ILR sharing to have better performance by just reusing a training configuration well tuned in individual learning. Its effectiveness on classification accuracy will be validated in Section 4.1.

**Balance between hard and soft loss objectives.** We follow the suggestion in [10] that the backpropagation flow from each soft objective should be multiplied by $T^2$ since the magnitudes of the gradients produced by the soft targets scale as $1/T^2$. This ensures that the relative contributions of the hard and soft targets remain roughly unchanged when tuning $T$.

## 3.4 Robustness to label noise

In supervised learning, it is hard to completely avoid confusion during network training either due to incorrect labels or data augmentation. For example, random cropping is a very important data augmentation technique when training an image classifier. However, the entire labeled objects or large portion of them occasionally get cut off, which really challenges the classifier. Since multiple views on the same example have diversity of predictions, collaborative learning is by nature more robust to label noise than individual learning, which will be validated in Section 4.1.

# 4 Experiments

We will evaluate the performance of collaborative learning on various network architectures for several datasets, with analysis of important and interesting observations. We use $T = 2$ and $\beta = 0.5$ for all experiments. In addition, the performance of any model trained with collaborative learning is evaluated using the first classifier head without head selection. All experiments are conducted with Tensorflow [1].

## 4.1 CIFAR Datasets

The two CIFAR datasets, CIFAR-10 and CIFAR-100, consist of colored natural images with 32x32 pixels [13] and have 10 and 100 classes, respectively. We conduct empirical studies on the CIFAR-10 dataset with ResNet-32, ResNet-110 [9], and DenseNet-40-12 [11]. ResNets and DenseNets for CIFAR are all designed to have three building blocks, residual or dense blocks. For the simple ILR sharing, the split point is just after the first block. For the hierarchical sharing, the two split points are located after the first and second blocks, respectively. Refer to Section 2 in Supplementary material for the detailed training setup.

Table 1: **Test errors (%) on CIFAR-10.** All experiments are performed 5 runs except for those of DenseNet-40-12 are done for 3 runs.

|  |  | ResNet-32 | ResNet-110 | DenseNet-40-12 |
|---|---|---|---|---|
| Individual learning | Single instance | $6.66 \pm 0.21$ | $5.56 \pm 0.16$ | $5.26 \pm 0.08$ |
|  | Label smoothing (0.05) | $6.83 \pm 0.14$ | $5.66 \pm 0.08$ | $5.40 \pm 0.04$ |
| Collaborative learning | 2 instances | $6.19 \pm 0.17$ | $5.21 \pm 0.14$ | $5.11 \pm 0.15$ |
|  | 4 instances | $6.16 \pm 0.17$ | $5.16 \pm 0.13$ | $5.00 \pm 0.05$ |
|  | 2 heads w/ simple ILR sharing | $5.97 \pm 0.07$ | $5.15 \pm 0.14$ | $5.04 \pm 0.10$ |
|  | 4 heads w/ hierarchical ILR sharing | $\mathbf{5.86} \pm 0.13$ | $\mathbf{4.98} \pm 0.12$ | $\mathbf{4.86} \pm 0.12$ |

**Classification results.** All results are summarized in Table 1. It can be concluded from Table 1 that with a given training graph pattern, the more classifier heads, the lower generalization error. More important, ILR sharing reduces not only GPU memory consumption and training time but also the generalization error considerably.

**Simultaneous vs alternative optimization.** We repeat an experiment that was performed in [23]. It is just a special case of collaborative learning in which we train two instances of ResNet-32 on CIFAR-100 with $T = 1, \beta = 0.5$. The only difference is that we replace the alternative optimization [23] with the simultaneous one. It is shown in Table 2 that based on the corresponding baseline, simultaneous optimization provides additional 1%+ accuracy gain compared to alternative one. With $T = 2$, simultaneous one has another 1% boost. Thus, simultaneous optimization substantially outperforms alternative one in terms of accuracy and speed.

Table 2: Alternative optimization [23] vs simultaneous optimization (ours) in terms of test errors of ResNet-32 on CIFAR-100.

|  |  | Single instance (baseline) | Head 1 in two instances | Head 2 in two instances |
|---|---|---|---|---|
| [23] |  | 31.01 | 28.81 | 29.25 |
| Collaborative learning | T=1 | $30.52 \pm 0.35$ | $27.48 \pm 0.37$ | $27.64 \pm 0.36$ |
|  | T=2 |  | $\mathbf{26.36} \pm 0.27$ | $\mathbf{26.32} \pm 0.26$ |

**Backpropagation rescaling.** Backpropagation rescaling is proposed to be necessary for ILR sharing theoretically in Section 3.3. We intend to confirm it by experiments on the CIFAR-10 dataset. To train a ResNet-32, we use a simple ILR sharing topology with four heads, and the split point located after the first residual block. The results in Table 3 provide evidence that backpropagation rescaling clearly outperforms others – no scaling and loss scaling. While no scaling suffers from too large gradients in the shared layers, loss scaling results in a too small factor for updating the parameters

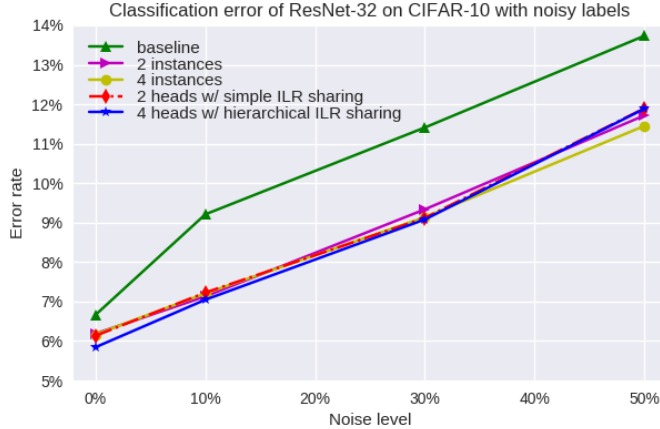

Figure 3: **Test error on CIFAR-10 with label noise**. Noise level is the percentage of corrupted labels over the all training set. The noisy labels are randomly generated every epoch.

of independent layers. We suggest backpropagation rescaling for all multi-head learning problems beyond collaborative learning.

Table 3: **Impact of backprop rescaling.** Four heads based on ResNet-32 share the low layers up to the first residual block. With no scaling, the factor for each head's loss is one. With loss scaling, the factor for each head's loss is 1/4.

|  | No scaling | Loss scaling | Backprop rescaling |
|---|---|---|---|
| Error (%) of ResNet-32 | $6.04 \pm 0.17$ | $6.09 \pm 0.24$ | $\mathbf{5.82} \pm 0.08$ |

**Noisy label robustness.** In this experiment, we aim at validating the noisy label resistance of collaborative learning on the CIFAR-10 dataset with ResNet-32. Assume that a portion of labels, whose percentage is called noise level, are corrupted with a uniform distribution over the label set. The partition for images with corruption or not is fixed for all runs; their noisy labels are randomly generated every epoch. The results in Figure 3 validate that the test error rates of all collaborative learning setups are substantially lower than the baseline, and the accuracy gain becomes larger at a considerably larger noise level. It is well expected since the consensus formed from a group is able to mitigate the effect of noisy labels without knowledge of noise distribution. Another observation is that 4 heads with hierarchical ILR sharing, which constantly provides the lowest error rate at a relatively low noise level, seems worse at a high noise level. We conjecture that the diversity of predictions is more important than better ILR sharing in this scenario. Collaborative learning provides flexibility to trade off the diversity of predictions from the group with additional supervision and regularization for the common layers.

## 4.2 ImageNet Dataset

The ILSVRC 2012 classification dataset consists of 1.2 million for training, and 50,000 for validation [6]. We evaluate how collaborative learning helps improve the performance of ResNet-50 network. As following the notations in [9], we consider two heads sharing ILRs up to "conv3_x" block for simple ILR sharing. For the hierarchical sharing with four heads, two split points are located after "conv3_x" and "conv4_x" blocks, respectively. Refer to Section 3 in Supplementary material for the detailed training setup.

**Classification error vs training computing resources (GPU memory consumption as well as training time).** Classification error on Imagenet is particularly important because many state-of-the-art computer vision problems derive image features or architectures from ImageNet classification models. For instance, a more accurate classifier typically leads to a better object detection model based on the classifier [12]. Table 4 summarizes the performance of various training graph patterns

Table 4: **Validation errors of ResNet-50 on ImageNet**. Label smoothing, distillation and collaborative learning all do not affect inference's memory size and running time.

|  |  | Top-1 error | Top-5 error | Training time | Memory |
|---|---|---|---|---|---|
| Individual learning | Baseline | 23.47 | 6.83 | 1x | 1x |
|  | Label smoothing (0.1) | 23.34 | 6.80 | 1x | 1x |
| Distillation | From ensemble of two ResNet-50s | 22.65 | 6.34 | 3.42x | 1.05x |
| Collaborative learning | 2 instances | 22.81 | 6.45 | 2x | 2x |
|  | 2 heads w/ simple ILR sharing | 22.70 | 6.37 | 1.4x | 1.32x |
|  | 4 heads w/ hierarchical ILR sharing | **22.29** | **6.21** | 1.75x | 1.5x |

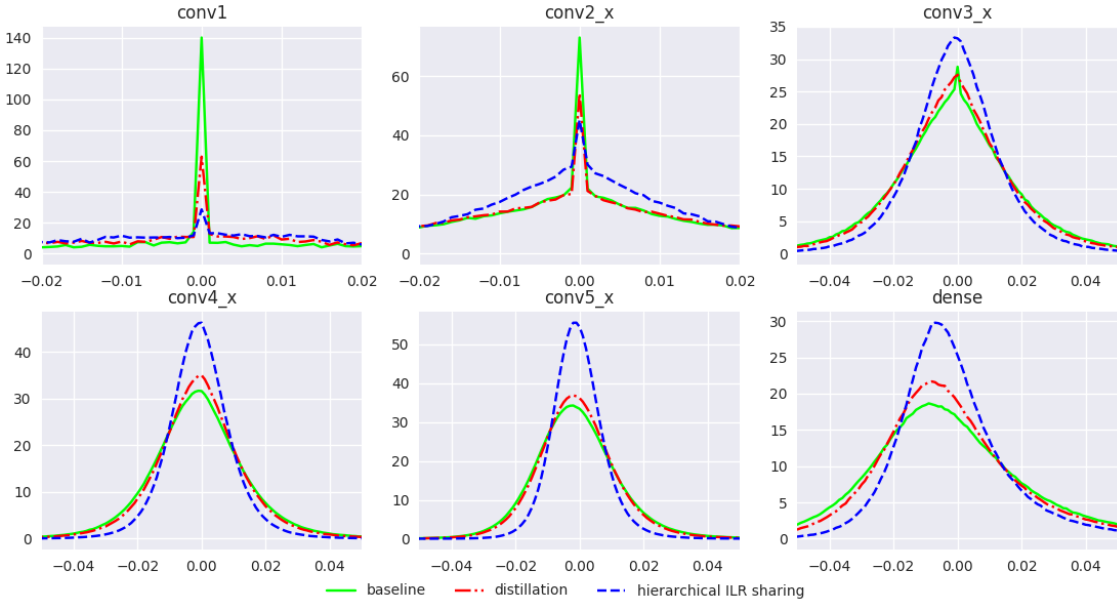

Figure 4: **Per-layer weight distribution in trained ResNet-50**. As following the notations in [9], the two split points in the hierarchical sharing with four heads are located after "conv3_x" and "conv4_x" blocks, respectively.

with ResNet-50 on ImageNet. As mentioned in Section 3.1, collaborative learning brings some extra training cost since it generates more classifier heads in training, and ILR sharing is designed for training speedup and memory consumption reduction. We have measured GPU memory consumption and training time and also listed them in Table 4. It is similar to the CIFAR results that two heads with simple ILR sharing and four heads with hierarchical ILR sharing reduce the validation top-1 error rate significantly in this case, from 23.47% with the baseline to 22.70% and 22.29%, respectively. Note that increasing training time for individual learning does not improve accuracy [22]. Since the convolution filters are shared in the space domain in deep convolutional networks, the memory consumption by storing the intermediate feature maps is much higher than that by model parameters in training [17]. Therefore, ILR sharing is especially computationally efficient for deep convolutional networks because it contains only one copy of shared layers. Compared to distillation[1], collaborative learning can achieve a lower error rate with a much less training time in an end-to-end way.

**Model weight distribution and mechanisms of ILR sharing.** We have plotted the statistical distribution of each layer's weights of trained ResNet-50 in Figure 4, including the baseline, distilled and trained versions with hierarchical ILR sharing. Refer to Section 5 in Supplementary material for more results with other training configurations. The first finding is that the weight distribution of the baseline has a very large spike at near zero in the bottom layers. We conjecture that the gradients to

many weights may be vanished so small that the weight decay part takes the major impact, which causes near-zero "dead" values eventually[2]. Compared to distillation, ILR sharing more effectively helps reduce the number of "dead" weights, thereby improve the accuracy. The second finding is that collaborative learning makes the weight distribution be more centralized to zero overall. Note that we also calculate per-layer model weight standard deviation values in Table 1 in Supplementary material to additionally support this claim. The results indicate that the consensus of multiple views on the same data provides additional regularization.

ILR sharing is somewhat related to the concept of hint training [18], in which a teacher transfers its knowledge to a student network by using not only the teacher's predictions but also an ILR. In collaborative learning, ILR sharing can be regarded as an extreme case in which the ILRs of two separated classifier heads converge to the exact same one by forcing them to match. It is reported in [18] that using hints can outperform distillation. To a certain extent, this provides an indirect evidence for the possibility of accuracy improvement from ILR sharing.

Again, two hyperparameters $\beta$ and $T$ are fixed in all of our experiments. It is possible that more extensive hyper-parameter searches may further improve the performance on specific datasets. We evaluate the impact of hyperparameters, $\beta$, $T$, and split point locations for ResNet-32 on CIFAR-10 in Section 6 in Supplementary material.

## 5   Conclusion

We have proposed a framework of collaborative learning to train a deep neural network in a group of generated classifiers based on the target network. The consensus of multiple views from different classifier heads on the same example provides supplementary information as well as regularization to each classifier, thereby improving the generalization. By well aggregating the gradient flows from all heads, ILR sharing with backpropagation rescaling not only lowers training computational cost, but also facilitates supervision to the shared layers. Empirical results have also validated the advantages of simultaneous optimization and backpropagation rescaling in group learning. Overall, collaborative learning provides a flexible and powerful end-to-end training approach for deep neural networks to achieve better performance. Collaborative learning also opens up several possibilities for future work. The mechanism of group collaboration and noisy label resistance imply that it may potentially be beneficial to semi-supervised learning. Furthermore, other machine learning tasks, such as regression, may take advantage of collaborative learning as well.

## Acknowledgement

We would like to thank Qiqi Yan for many helpful discussions.

## Footnotes

[1]Training time of distillation is analyzed in Section 4 in Supplementary material.

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
