[Supplementary Material]

# Supplementary Material of Collective Training for Deep Neural Networks

**Guocong Song**
Playground Global
Palo Alto, CA 94306
songgc@gmail.com

**Wei Chai**
Google
Mountain View, CA 94043
chaiwei@google.com

## 1 Proof of variance of gradient with backpropagation rescaling being finite for an arbitrary number of heads $H$

This is equivalent to prove that $\text{Var}\left(\frac{1}{H}\sum_{h=1}^{H}X_h\right) < \infty$ for all $H$ if $\text{Var}(X_h) < \infty$ for $\forall h$.

*Proof.*

$$\text{Var}\left(\frac{1}{H}\sum_{h=1}^{H}X_h\right) = \sum_{h=1}^{H}\text{Var}(X_h/H) + \sum_{i\neq j}\text{Cov}(X_i/H, X_j/H) \tag{1}$$

$$\leq \frac{1}{H^2}\sum_{h=1}^{H}\text{Var}(X_h) + \frac{1}{H^2}\sum_{i\neq j}|\text{Cov}(X_i, X_j)| \tag{2}$$

$$\leq \frac{1}{H^2}\sum_{h=1}^{H}\text{Var}(X_h) + \frac{1}{H^2}\sum_{i\neq j}\sqrt{\text{Var}(X_i)}\sqrt{\text{Var}(X_j)} \tag{3}$$

$$\leq \frac{1}{H^2}H^2\max_h(\text{Var}(X_h)) \tag{4}$$

$$= \max_h(\text{Var}(X_h)) \tag{5}$$

Inequality (3) is because of Cauchy–Schwarz inequality. Therefore, if $\text{Var}(X_h) < \infty$ for $\forall h$, $\text{Var}\left(\frac{1}{H}\sum_{h=1}^{H}X_h\right) < \infty$ as well. $\qquad\square$

## 2 Training setup for CIFAR

We adopt a standard data augmentation scheme that is widely used for those two datasets [2, 3]. We train three target networks: ResNet-32, ResNet-110 [2], and DenseNet-40-12 [4]. In training, we use a weight decay of $10^{-4}$, and a Nesterov momentum of 0.9 for SGD for all networks. ResNet-32 and ResNet-110 are trained with a mini-batch size of 128 up to 200 epochs. For them, we start with a learning rate of 0.1 and divide it by 10 at 100, 150, and 192 epochs. DenseNet-40-12 is trained with a mini-batch size of 64 up to 300 epochs. Its learning rate is initially set to 0.1 and is divided it by 10 at 150, 225, and 290 epochs.

## 3 Training setup for ImageNet

We adopt the same data augmentation scheme for training images as in [1]. Each network input image is a 224x224 pixel random crop from an augmented image or its horizontal flip, and then is

normalized by the per-color mean and standard deviation. We train ResNet-50 [2] with a Nesterov momentum [5] of 0.9 and a weight decay of $10^{-4}$ up to 100 epochs. Each GPU consumes 32 images per mini-batch. The learning rate is initially set to 0.1, and then is divided by 10 at 30, 60, and 90 epochs. A single central crop with size of 224x224 is applied for validation.

# 4  Training time of distillation

The training time of distillation can be expressed as

$$T_{train} = T_t + T_s + T_{tf}$$

where $T_t$ is the training time of the teacher network, $T_s$ is that of the student one, and $T_{tf}$ is the forward passing time of the teacher during distillation. For example, when distilling a ResNet-50 from an ensemble of two ResNet-50s. $T_t = 2T_s$, and $T_{tf} \approx 0.4T_s$. Therefore, the total training time is roughly 3.4x that with individual learning.

# 5  Details of ResNet-50 weight distribution

The distributions in other cases are shown in Figure 1. Per-layer weight standard deviation values are listed with different training approaches in Table 1.

To validate our conjecture that the gradients to many weights in the bottom layers may be vanished so small that the weight decay part takes the major impact, which causes near-zero "dead" values eventually, we perform an experiment in which the value of weight decay is reduced to $0.5 \cdot 10^{-4}$ in conv1, conv2_x, and conv3_x layers, and that in other layers remains to be $1 \cdot 10^{-4}$. Figure 2 shows the expected results that a reduced weight decay does reduce the spike in the weight distribution. However, it does not reduce the error rate of the classifier, which is 23.5% for the top-1 error. Therefore, although weight decay is related to these "dead" filter weights, simply reducing weight decay is not a solution to improve accuracy.

Table 1: Per-layer weight standard deviation in ResNet-50

|  |  | conv1 | conv2_x | conv3_x | conv4_x | conv5_x | dense |
|---|---|---|---|---|---|---|---|
| Individual learning | Baseline | 0.116 | 0.034 | 0.024 | 0.017 | 0.014 | 0.033 |
|  | Label smoothing (0.1) | 0.103 | 0.029 | 0.021 | 0.015 | 0.013 | 0.027 |
| Distillation | From ensemble of two ResNet-50s | 0.113 | 0.035 | 0.022 | 0.016 | 0.013 | 0.030 |
| Collaborative learning | 2 instances | 0.077 | 0.024 | 0.016 | 0.011 | 0.009 | 0.022 |
|  | 2 heads w/ simple ILR sharing | 0.078 | 0.025 | 0.017 | 0.011 | 0.009 | 0.022 |
|  | 4 heads w/ hierarchical ILR sharing | 0.076 | 0.024 | 0.016 | 0.011 | 0.008 | 0.022 |

# 6  Impact of hyperparameters on accuracy on CIFAR-10

## 6.1  Impact of $\beta$ and $T$

We have run some experiments with different $\beta$ and $T$ values and plotted the results in Fig 3. The error is not sensitive to them. Carefully tuning $\beta$ and $T$ could obtain better results from the current settings ($\beta = 0.5$, $T = 2$), but the improvement is expected to be small.

## 6.2  Impact of split point location

We evaluate the impact of different split point locations in ResNet-32 with 2-head simple ILR sharing on CIFAR-10, and summarize the results in Table 2.

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

Figure 2: **Per-layer weight distribution in trained ResNet-50 with baseline and per-layer weight decay**. For the baseline, the value of weight decay is $1 \cdot 10^{-4}$. For per-layer weight decay, the value of weight decay is $0.5 \cdot 10^{-4}$ for conv1, conv2_x, and conv3_x layers, and $1 \cdot 10^{-4}$ otherwise. However, its top-1 error rate is 23.5%, and not improved from the baseline.

Table 2: **Error of ResNet-32 on CIFAR-10 with different split point locations.** Simple ILR sharing is applied with two heads. RB is short for residual block.

|  | Before RB 1 | After RB 1 | After RB 2 | After RB 3 |
|---|---|---|---|---|
| Error (%) | $6.25 \pm 0.16$ | $\mathbf{5.97 \pm 0.07}$ | $6.49 \pm 0.10$ | $6.68 \pm 0.11$ |

Figure 3: Error of CIFAR-10 using Resnet-32 with hierarchical ILR sharing