[Reviews · NeurIPS 2018]

Reviewer 1



Summary The paper proposes a collaborative learning strategy in which multiple classifier heads of the same deep network are trained simultaneously on the same data. Unlike ensembling, knowledge distillation, multi-task learning or training auxiliary classifiers, this seems more effective from the point of view of training time and memory utilization. In addition, the proposed training objective improves generalization and makes the network more robust to label noise. For the intermediate representations that are shared, the authors propose a backpropagation-rescaling technique so as to control the variance of gradients being propagated to the previous layers. Empirical results demonstrate the efficacy of the method w.r.t. Robustness to label noise and generalization improvements over other approaches and baselines. Strengths - The paper is clear, well-written and easy to follow. Several design choices - the formulation of the objective function, the gradient re-scaling, etc. are well-motivated and demonstrated via empirical results. The proposed approach in itself is simple and seems easy to implement within limited additional memory capacity. - The paper presents a fresh-perspective on the problem of improving generalization and robustness to noise. - The results presented in Table. 4 and Figure. 3 show clear improvements over the baselines. Backpropagation scaling clearly leads to better results (Table. 3). Weaknesses - The authors mention they fixed the hyper-parameters \beta and T in all their experiments. Were these hyper-parameters chosen by cross-validation? If not, how sensitive are the results to the choice of \beta and T. I am curious if some sort of annealing schedule for \beta and T leads to better results. ----------- I read all the reviews and the feedback provided by the authors. The rebuttal addressed my concern regarding the choice of the hyper-parameters and annealing schedule for \beta and T with experimental results. Given from what I understood from the paper, feedback and some of the comments mentioned by the other reviewers, I am inclined towards sticking to my original score regarding the paper.

Reviewer 2



A collaborative learning framework for training deep neural networks is presented in the paper, targeting at improved accuracy. It is an interesting formulation, starting with a rather simple operation, based on using multiple classifier heads in the training procedure. Although significance of outcomes should be better determined, the formulation and optimization procedure described in Section 3 show a good potential for the use of the methodology. Effectively determining the hyperparameter values is an issue to consider. A somewhat better evaluation of the Table 4 results (training time vs memory wrt accuracy) could be presented.

Reviewer 3



The paper introduces collaborative learning, a modification to architectures that adds additional heads to the classifier in turn enabling better generalization without added inference costs. Strengths: - The topic of study is of great relevance and significance. While ensembling of networks boosts performance, the added inference cost is typically not warranted; having a way to distill ensemble-like performance in a single model is impactful. - The authors present results on CIFAR and ImageNet datasets which show how the proposed heuristics improve performance. The ablation experiments are useful and add to the reader's understanding of the changes. Weaknesses: - The idea of using heads (or branches) in deep networks is not entirely novel and the authors do not describe or compare against past work such as mixture of experts and mixture of softmax (ICLR 2018). The authors should also investigate the Born Again Networks (ICML 2018) work that uses self-distillation to obtain improved results. It would also be beneficial if the authors described auxiliary training in the related work since they explain their approach using that as foundation (Section 3.1). - The proposed changes can be explained better. For instance, it is not clear how the hierarchy should be defined for a general classifier and which of the heads is retained during inference. - What baseline is chosen in Figure 3? Is it also trained on a combination of hard and soft targets? - The approach presented is quite heuristic it would be desirable if the authors could discuss some theoretical grounding for the proposed ideas. This is especially true for the section on "backpropagation rescaling". Did the authors try using a function that splits the outputs equally for all heads (i.e., I(x) = x/H, I'(x) = 1/H)? The argument for not using sum_h Lh is also not convincing. Why not modulate eta accordingly? - The paper would benefit from careful editing; there are several typos such as "foucus", "cross entroy", and awkward or colloquial phrases such as "requires to design different classifier heads", "ILR sharing with backpropagation rescaling well aggregates the gradient flow", "confusion often happens to network training", "put on top of individual learning", "SGD has nicer convergence". Post-rebuttal: Raising my score to a 6 in response to the rebuttal. I do wish to point out that: - To my understanding, BANs don't ensemble. They only self-distill. - Is using Lhard the right baseline for Q3?